# BlackMarks: Black-box Multi-bit Watermarking for Deep Neural Networks

## Abstract

Deep Neural Networks (DNNs) are increasingly deployed in cloud servers and autonomous agents due to their superior performance. The deployed DNN is either leveraged in a white-box setting (model internals are publicly known) or a black-box setting (only model outputs are known) depending on the application. A practical concern in the rush to adopt DNNs is protecting the models against Intellectual Property (IP) infringement. We propose BlackMarks, the first end-to-end multi-bit watermarking framework that is applicable in the black-box scenario. BlackMarks takes the pre-trained unmarked model and the owner's binary signature as inputs. The output is the corresponding marked model with specific keys that can be later used to trigger the embedded watermark. To do so, BlackMarks first designs a model-dependent encoding scheme that maps all possible classes in the task to bit '0' and bit '1'. Given the owner's watermark signature (a binary string), a set of key image and label pairs is designed using targeted adversarial attacks. The watermark (WM) is then encoded in the distribution of output activations of the DNN by fine-tuning the model with a WM-specific regularized loss. To extract the WM, BlackMarks queries the model with the WM key images and decodes the owner's signature from the corresponding predictions using the designed encoding scheme. We perform a comprehensive evaluation of Black-Marks' performance on MNIST, CIFAR-10, ImageNet datasets and corroborate its effectiveness and robustness. BlackMarks preserves the functionality of the original DNN and incurs negligible WM embedding overhead as low as $2.054\%$.

## 1 Introduction

Deep neural networks and other Deep Learning (DL) variants have revolutionized various critical fields ranging from biomedical diagnosis and autonomous transportation to computer vision and natural language processing (Deng & Yu, 2014; Ribeiro et al., 2015). Training a highly accurate DNN is a costly process since it requires: (i) processing massive amounts of data acquired for the target application; (ii) allocating substantial computing resources to fine-tune the underlying topology (i.e., type and number of hidden layers), and hyper-parameters (i.e., learning rate, batch size), and DNN weights to obtain the most accurate model. Therefore, developing a high-performance DNN is impractical for the majority of customers with constrained computational capabilities. Given the costly process of designing/training, DNNs are typically considered to be the intellectual property of the model builder and needs to be protected to preserve the owner's competitive advantage.

Digital watermarking has been immensely leveraged over the past decade for ownership protection in the multimedia domain where the host of the watermark can be images, video contents, and functional artifacts such as digital integrated circuits (Furht & Kirovski, 2004; Hartung & Kutter, 1999; Qu & Potkonjak, 2007). However, the development of DNN watermarking techniques is still in its early stage. Designing a coherent DNN watermarking scheme for model ownership proof is challenging since the embedded WM is required to yield high detection rates and withstand potential attacks while minimally affecting the original functionality and overhead of the target DNN.

Existing DNN watermarking techniques can be categorized into two types depending on the application scenario. 'White-box' watermarking assumes the availability of model internals (e.g., weights) for WM extraction (Uchida et al., 2017) whereas 'black-box' watermarking assumes that the output predictions can be obtained for WM detection (Merrer et al., 2017; Adi et al., 2018b). On the one hand, white-box WMs have a larger capacity (carrying multiple-bit information) but limited appli-

cability due to the strong assumption. On the other hand, black-box WMs enable IP protection for Machine Learning as a Service (MLaaS) (Ribeiro et al., 2015) where only zero-bit watermarking methods have been proposed. It is desirable to develop a systematic watermarking approach that combines the advantages of both types of WMs. While all present black-box watermarking papers embed the WM as a statistical bias in the decision boundaries of the DNN (high accuracy on the WM trigger set), our work is the first to prove that it is feasible to leverage the model's predictions to carry a multi-bit string instead of a one-bit boolean decision (existence or not of the WM).

By introducing BlackMarks, this paper makes the following contributions:

- Proposing BlackMarks, the first end-to-end black-box watermarking framework that enables multi-bit WM embedding. BlackMarks possesses higher capacity compared to prior works and only requires the predictions of the queried model for WM extraction.
- Characterizing the requirements for an effective watermarking methodology in the deep learning domain. Such metrics provide new perspectives for model designers and enable coherent comparison of current and pending DNN IP protection techniques.
- Performing extensive evaluation of BlackMarks' performance on various benchmarks. Experimental results show that BlackMarks enables robust WM embedding with high detection rates and low false alarm rates. As a side benefit, we find out that BlackMarks' WM embedding process improves the robustness of the model against adversarial attacks.

## 2 PRELIMINARY AND RELATED WORK

Digital watermarks are invisible identifiers embedded as an integral part of the host design and have been widely adopted in the multimedia domain for IP protection (Lu, 2004; Cox et al., 2007). Conventional digital watermarking techniques consist of two phases: WM embedding and WM extraction. Figure 1 shows the workflow of a typical constraint-based watermarking system. The original problem is used as the cover constraints to hide the owner's WM signature. To embed the WM, the IP designer creates the stego-key and a set of additional constraints that do not conflict with cover constraints. Combining these two constraints yields the stego-problem, which is solved to produce the stego-solution. Note that the stego-solution simultaneously satisfies both the original constraints and the WM-specific constraints, thus enables the designer to extract the WM and claim the authorship. An effective watermarking method is required to meet a set of criteria including imperceptibility, robustness, verifiability, capacity, and low overhead (Singh & Chadha, 2013).

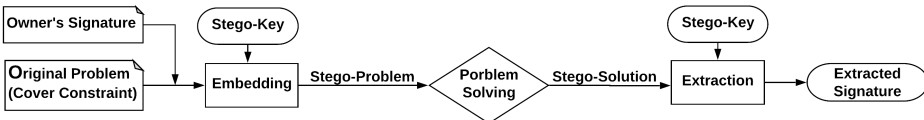

Figure 1: Illustration of a typical constraint-based watermarking system.

IP protection of valuable DNN models is a subject of increasing interests to researchers and practitioners. Uchida et al. (2017) take the first step towards DNN watermarking by embedding the WM in the weights of intermediate layers and training the model with an additional regularization loss. The WM is later extracted from the weights of the marked layer assuming a white-box scenario. Rouhani et al. (2018a) present the first generic watermarking approach that is applicable in both white-box and black-box settings by embedding the WM in the activation maps of the intermediate layers and the output layer, respectively. To alleviate the constraint on the availability of model internals during WM extraction, several papers propose zero-bit watermarking techniques that are applicable in the black-box scenario. Merrer et al. (2017) craft adversarial samples to carry the 'zero-bit' WM and embeds them in the decision boundary of the model by fine-tuning the DNN with the WM. Null hypothesis testing is performed to detect the WM based on the remote model's response to the WM query images. Adi et al. (2018a) suggest to use the incorrectly classified images from the training data as the WM trigger images and generate random labels as the corresponding key labels. A commitment scheme is applied to the trigger set to produce the WM marking key and the verification key. The existence of the WM is determined by querying the model with the marking keys and performing statistical hypothesis testing. Zhang et al. (2018) propose three WM generation algorithms ('unrelated', 'content', 'noise') and embed the WM by training the model with the concatenation of the training set and the WM set. To detect the WM, the remote model is queried by the WM set and the corresponding accuracy is thresholded to make the binary decision. To the best of our knowledge, none of the prior works has addressed the problem of multi-bit black-box watermarking.

# 3 BLACKMARKS OVERVIEW

Figure 2 shows the global flow of BlackMarks framework. BlackMarks consists of two main phases: WM embedding and WM extraction. The workflow of each phase is explained below.

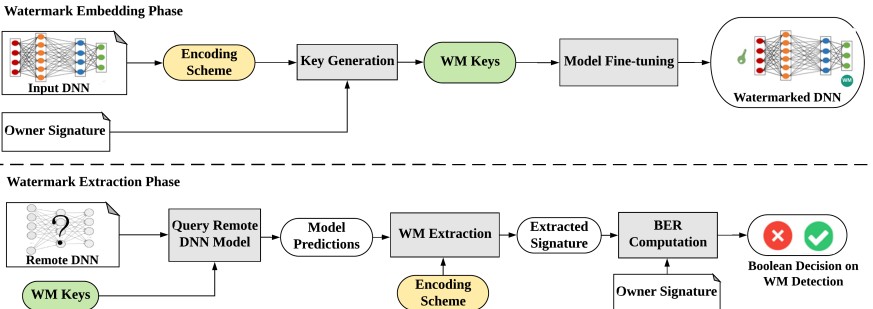

Figure 2: BlackMarks' Global Flow.

**Watermark Embedding.** The WM embedding module of BlackMarks takes the pre-trained model and the owner-specific WM signature as its inputs. The output is the watermarked DNN together with a set of WM keys. A model-dependent encoding scheme is devised to map all possible labels into bit '0' and bit '1'. Note that WM embedding is a one-time task performed locally by the owner before the model is distributed. Details of each step are discussed in Section 4.1.

**Watermark Extraction.** To verify the IP of a remote DNN and detect potential IP infringement, BlackMarks first queries the DNN service with the WM keys. The owner's signature is then decoded from the corresponding predictions using the encoding scheme. The Bit Error Rate (BER) between the extracted signature and the true one is computed. A zero BER implies that the owner's IP is deployed in the remote DNN service. Section 4.2 discusses the details of WM extraction.

## 3.1 EVALUATION CRITERIA

Table 1 details the evaluation criteria for an effective DNN watermarking methodology. In addition to previously suggested requirements in Uchida et al. (2017); Merrer et al. (2017), we believe verifiability and integrity are two other major factors that need to be considered when designing a practical DNN watermarking methodology. *Verifiabiity* is important because the embedded signature should be accurately extracted using the pertinent WM keys; the model owner is thereby able to detect any misuse of her model with a high probability. *Integrity* ensures that the IP infringement detection policy yields a minimal false alarms rate, meaning that there is a very low chance of falsely proving the ownership of an unmarked model used by a third party. BlackMarks satisfies all the requirements listed in Table 1 as we empirically show in Section 5.

Table 1: Evaluation criteria for an effective watermarking of deep neural networks.

| Requirements | Description |
|---|---|
| Fidelity | Accuracy of the target neural network shall not be degraded as a result of watermark embedding. |
| Verifiability | Watermark extraction shall yield minimal false negatives; WM shall be effectively detected using the pertinent keys. |
| Robustness | Embedded watermark shall withstand model modifications such as pruning, fine-tuning, or WM overwriting. |
| Integrity | The authorship of the unmarked models will not be falsely claimed by the watermarking method. |
| Capacity | Watermarking methodology shall be capable of embedding a large amount of information in the target DNN. |
| Efficiency | Communication and computational overhead of watermark embedding and extraction shall be negligible. |
| Security | The watermark shall be secure against brute-force attacks and leave no tangible footprints in the target DNN. |

## 3.2 ATTACK MODEL

To validate the robustness of a potential DL watermarking approach, one should evaluate the performance of the proposed methodology against (at least) three types of contemporary attacks: (i) **model fine-tuning**. This attack involves re-training of the original model to alter the model parameters and find a new local minimum while preserving the accuracy. (ii) **parameter pruning**. Pruning is a common approach for efficient DNN execution, particularly on embedded devices. We identify model pruning as another attack approach that might affect the watermark extraction. (iii) **watermark overwriting**. A third-party user who is aware of the methodology used for DNN watermarking may try to embed a new WM in the distributed model and overwrite the original one.

## 4 BLACKMARKS METHODOLOGY

BlackMarks takes advantage of the fact that there is not a unique solution for modern non-convex optimization problems used by DL models (Choromanska et al., 2015; Rouhani et al., 2017) to embed the WM signature in the output activations of the target DNN. In this section, we detail the workflow of WM embedding and extraction phase shown in Figure 2 and discuss the watermarking overhead. We use image classification as the cover problem in this paper, however, BlackMarks can be easily generalized to other data applications for IP protection of the deployed models.

### 4.1 WATERMARK EMBEDDING

Algorithm 1 summarizes the steps of BlackMarks' WM embedding. BlackMarks encodes the owner-specific WM information in the distribution of the output activations (before *softmax*) while preserving the correct functionality on the original task. The rationale behind our approach is to explore the unused space within the high dimensional DNN (Rouhani et al., 2018b) for WM embedding. Black-Marks formulates WM embedding as a one-time, *'post-processing'* step that is performed on the pre-trained DNN locally by the owner before model distribution/deployment. We explicitly discuss each of the steps outlined in Algorithm 1 in the following of this section.

---

**Algorithm 1** BlackSigns's black-box multi-bit watermark embedding framework.

**INPUT: Target model** $(\mathcal{M})$**; Training data** $(\{X^{train}, Y^{train}\})$**; WM Key length** $(K)$**, Owner's signature** $\mathbf{b} \in \{0,1\}^K$**, Total number of classes** $(C)$

**OUTPUT: Watermarked DNN** $(\mathcal{M}^*)$**; Crafted WM keys** $(\{X^{key}, Y^{key}\})$**, Encoding scheme**(f)

**1** Encoding Scheme Design:
 **for** i in range(C): **do**
  $A_i \leftarrow Get\_Activations(\ \mathcal{M},\ \{X^{train}, Y^{train}\}, i);\ a_i = Get\_Mean(A, axis = 0)$
 $\mathbf{f} = [bit0, bit1] \leftarrow K\_Means\_Clustering(\ [a_0, ..., a_{C-1}],\ num\_components = 2)$

**2** Key Generation 1:
 Set the initial key size $K' > K$ (e.g., $K' = 10 \times K$).
 $\{X^{key'}, Y^{key'}\} \leftarrow Generate\_WM\_Keys(\mathcal{M}, \{X^{train}, Y^{train}\}, \mathbf{f}, K')$

**3** Model Fine-tuning: The additive regularization loss is incorporated to train the DNN:
 $\mathcal{M}^* \leftarrow Train(\mathcal{M}, \{X^{key'}, Y^{key'}\}, \{X^{train}, Y^{train}\}, \mathcal{L}_R)$

**4** Key Generation 2:
 $Y_{\mathcal{M}^*}^{pred'} \leftarrow Predict(\mathcal{M}^*, X^{key'});\ I_{\mathcal{M}^*} \leftarrow Get\_Match\_Index\ (Y^{key'}, Y_{\mathcal{M}^*}^{pred'})$
 $Y_{\mathcal{M}}^{pred'} \leftarrow Predict(\mathcal{M}, X^{key'});\ I_{\mathcal{M}} \leftarrow Get\_Mismatch\_Idx\ (Y^{key'}, Y_{\mathcal{M}}^{pred'})$
 Construct $T$ unmarked models: $\mathcal{M}_j = Train(\mathcal{M}, \{X^{train}, Y^{train}\}, \mathcal{L}_0),$ j=0,..., T-1
 **for** j in range(T) **do**:
  $Y_{\mathcal{M}_j}^{pred'} \leftarrow Predict(\mathcal{M}_j, X^{key'});\ I_{\mathcal{M}_j} \leftarrow Get\_Mismatch\_Idx\ (Y^{key'}, Y_{\mathcal{M}_j}^{pred'})$
 $I_{\mathcal{M}} \leftarrow Get\_Intersection\ (\ I_{\mathcal{M}}, I_{\mathcal{M}_0}, ..., I_{\mathcal{M}_{T-1}});\ I^{key'} \leftarrow Get\_Intersection\ (\ I_{\mathcal{M}}, I_{\mathcal{M}^*})$
 $\{X^{key}, Y^{key}\} \leftarrow Select\ (\ \{X^{key'}, Y^{key'}\},\ I^{key'}, K)$

 **Return:** Marked DNN $\mathcal{M}^*$; WM key $\{X^{key}, Y^{key}\}$; Encoding Scheme $\mathbf{f}$.

---

Figure 3: BlackMarks' WM embedding algorithm.

**1** **Encoding Scheme Design.** Recall that our objective is to encode a binary string (owner's signature) into the predictions made by the DNN when queried by the WM keys. BlackMarks designs a model-dependent encoding scheme that maps the class predictions to binary bits. The encoding scheme (**f**) is obtained by clustering the output activations corresponding to all categories $(C)$ into two groups based on their similarity. To do so, BlackMarks computes the averaged output activations triggered by images in each class and divides them into two groups using K-means clustering. The resulting encoding scheme specifies the labels corresponding to bit '0' and bit '1', respectively.

**2** **Key Generation 1.** The key generation module takes the encoding scheme, the owner's private signature ($\mathbf{b} \in \{0,1\}^K$, where $K$ is also used as the key length), and the original training data as inputs. The output is a set of WM key image-label pairs for the target DNN. More specifically, BlackMarks deploys *targeted adversarial attacks* to craft the WM key images and labels. If the given bit in $\mathbf{b}$ is '0', the source class and the target class for the WM image ('adversarial sample') are determined by uniformly randomly selecting a class label that belongs to the cluster '0' and cluster '1' determined by the encoding scheme (**f**), respectively. The source class is used as the corresponding WM key label. The WM keys for bit '1' in $\mathbf{b}$ are generated in a similar way. We use targeted

Momentum Iterative Method (MIM, Dong et al., 2018) and Jacobian-based Saliency Map Approach (JSMA, Papernot et al., 2016b) in our experiments (see Appendix A.1). BlackMarks framework is generic and compatible with other targeted adversarial attack emthods for key generation.

BlackMarks aims to design specific WM key images as the queries for model authentication instead of crafting standard adversarial samples that are indistinguishable to human eyes to fool the DNN. However, BlackMarks' WM key images can be considered as a generalization of 'adversarial samples' with relaxed constraints on the perturbation level and a different objective. We assume that the WM signature and the key generation parameters (e.g., source and target classes, maximum distortion, step size, and the number of attack iterations) are secrets specified by the owner. It's worth noting that the *transferability* of adversarial samples (Papernot et al., 2016a; 2017) might lead to false positives of WM detection as shown in Merrer et al. (2017). To address this problem, we set the initial key size to be larger than the owner's desired value $K' > K$ and generate the WM keys accordingly ($K' = 5 \times K$ in our experiments). The intuition here is that we want to filter out the highly transferable WM keys that are located near the decision boundaries.

**3** **Model Fine-tuning.** To enable seamless encoding of the WM information, BlackMarks incorporates an additive WM-specific embedding loss ($\mathcal{L}_{WM}$) to the conventional cross-entropy loss ($\mathcal{L}_0$) during DNN fine-tuning where a mixture of the WM keys and (a subset of) the original training data is fed to the model. The formulation of the total regularized loss ($\mathcal{L}_R$) is given in Eq. (1) where the embedding strength $\lambda$ controls the contribution of the additive loss. Here, we use *Hamming Distance* as the loss function $\mathcal{L}_{WM}$ to measure the difference between the extracted signature (obtained by $decode\_predictions$) and the true signature $\mathbf{b}$.

$$\mathcal{L}_R = \mathcal{L}_0 + \lambda \cdot \mathcal{L}_{WM}(\ \mathbf{b},\ decode\_predictions(Y_{\mathcal{M}^*}^{key}, \mathbf{f})). \qquad (1)$$

Note that without the additional regularization loss ($\mathcal{L}_{WM}$), this retraining procedure resembles 'adversarial training' (Kurakin et al., 2016). All of the existing zero-bit black-box watermarking papers (Merrer et al., 2017; Zhang et al., 2018; Adi et al., 2018a) leverages 'adversarial training' for WM embedding to ensure that the marked model has a high classification accuracy on the WM trigger set. However, such an approach does not directly apply to multi-bit WM embedding where we care about the difference between the decoded signature and the original one instead of the difference between the received predictions and the WM key labels. BlackMarks identifies this inherent requirement of multi-bit watermarking and formulates an additive embedding loss ($\mathcal{L}_{WM}$) to encode the WM. The rationale behind our design is that, when queried by WM key images, an additional penalty shall be applied if the prediction of the marked model does not belong to the same code-bit cluster as the corresponding WM key label.

**4** **Key Generation 2.** Once the model is fine-tuned with the regularized loss in step 3, we first find out the indices of initial WM keys that are correctly classified by the watermarked model (denoted by $I_{\mathcal{M}^*}$). To identify and remove WM keys images with high transferability, we construct $T$ variants of the original unmarked model ($T = 3$ in our experiments) to find out the indices of common misclassified initial keys (denoted by $I_{\mathcal{M}}$). Finally, the intersection of $I_{\mathcal{M}^*}$ and $I_{\mathcal{M}}$ determines the indices of proper key candidates to carry the WM signature. A random subset of candidate WM keys is then selected as the final WM keys according to the owner's key size ($K$). In the global flow (Figure 2), we merge the two key generation steps into one module for simplicity.

## 4.2 WATERMARK EXTRACTION

To extract the signature from the remote DNN ($\mathcal{M}'$), the owner queries the model with the WM key images ($X^{key}$) generated in step 4 of WM embedding and obtains the corresponding predictions ($Y_{\mathcal{M}'}^{key}$). Each prediction is then decoded to a binary value using the encoding scheme ($\mathbf{f}$) designed in WM embedding. The decoding is repeated for all predictions on the WM keys and yields the recovered signature ($\mathbf{b}'$). Finally, the BER between the true signature ($\mathbf{b}$) and the extracted one ($\mathbf{b}'$) is computed. The owner can prove the authorship of the model if the BER is zero.

## 4.3 WATERMARKING OVERHEAD

Here, we analyze the computation and communication overhead of WM extraction. The runtime overhead of the one-time WM embedding is empirically studied in Section 5.6. For the remote DNN service provider, the computation overhead of WM extraction is equal to the cost of one forward pass of WM key images through the underlying DNN. For the model owner, the computation cost

consists of two parts: (i) decoding the prediction response $Y_{M'}^{key}$ to a binary vector by finding out which cluster ('0' or '1' in the encoding scheme $\mathbf{f}$) contains each prediction; and (ii) performing an element-wise comparison between the recovered signature ($\mathbf{b}'$) and the true one ($\mathbf{b}$) to compute the BER. In this case, the communication overhead is equal to the key length ($K$) multiplied by the sum of the input image dimension and one to submit the queries and read back the predicted labels.

## 5 EVALUATION

We evaluate BlackMarks' performance on various datasets including MNIST (LeCun et al., 1998), CIFAR10 (Krizhevsky & Hinton, 2009) and ImageNet (Deng et al., 2009), with three different neural network architectures. The experimental setup and the network architectures are given in Appendix A.1 and A.2, respectively. We explicitly evaluate BlackMarks' performance with respect to each requirement listed in Table 1 as follows. Empirical results prove that BlackMarks is effective and applicable across various datasets and DNN architectures.

### 5.1 FIDELITY

Fidelity requires that the accuracy of the target neural network shall not be significantly degraded after WM embedding. Table 2 compares the baseline DNN accuracy (Column 2) and the accuracy of marked models (Column 3 and 4) after WM embedding. As demonstrated, BlackMarks respects the fidelity requirement by simultaneously optimizing for the classification accuracy of the underlying model (the cross-entropy loss), as well as the additive WM-specific loss as discussed in Section 4.1. In some cases (e.g. WideResNet benchmark), we even observe a slight accuracy improvement compared to the baseline. This improvement is mainly due to the fact that the additive loss $\mathcal{L}_{WM}$ in Eq. (1) act as a regularizer during DNN training. Regularization, in turn, helps the model to mitigate over-fitting by inducing a small amount of noise to DNNs (Goodfellow et al., 2016).

Table 2: Accuracy comparison between the unmarked baseline and the marked model.

| Dataset | Baseline Accuracy | Accuracy of Marked Model | | |
|---------|-------------------|----------|----------|----------|
| MNIST | 99.44% | K = 20 | K = 30 | K = 50 |
| | | 99.44% | 99.46% | 99.42% |
| CIFAR10 | 92.09% | K = 20 | K = 30 | K = 50 |
| | | 92.10% | 92.28% | 92.19% |
| ImageNet (top-1) | 56.39% | K = 20 | K =30 | K =50 |
| | | 56.28% | 56.31% | 56.36% |

### 5.2 VERIFIABILITY AND ROBUSTNESS

BlackMarks enables robust DNN watermarking and reliably extracts the embedded WM for ownership verification. We evaluate the robustness of BlackMarks against three state-of-the-art removal attacks as discussed in Section 3.2. These attacks include parameter pruning (Han et al., 2015), model fine-tuning (Simonyan & Zisserman, 2014), and watermark overwriting (Uchida et al., 2017).

**Model Fine-tuning.** Fine-tuning is a type of transformation attack that a third-party user might use to remove the WM information. To perform such an attack, the adversary retrains the distributed marked model using the original training data with the conventional cross-entropy loss (excluding $\mathcal{L}_{WM}$). Table 6 in Appendix A.3 summarizes the impact of fine-tuning on the watermark detection rate across all benchmarks. As can be seen from the table, the WM signature embedded by BlackMarks framework can be successfully extracted with zero BER even after the model is fine-tuned for various numbers of epochs.

**Parameter Pruning.** We use the pruning approach proposed in Han et al. (2015) to sparsify the weights in the target watermarked DNN. To prune a specific layer, we first set $\alpha\%$ of the parameters that possess the smallest weight values to zero. The model is then sparsely fine-tuned using cross-entropy loss to compensate for the accuracy drop caused by pruning. Figure 4 demonstrates the impact of pruning on WM extraction. One can see that BlackMarks tolerates up to $95\%$, $80\%$, and $90\%$ parameter pruning for MNIST, CIFAR-10, and ImageNet benchmark, respectively. As illustrated in Figure 4, in cases where DNN pruning yields a substantial BER value, the sparse model suffers from a large accuracy drop. Therefore, one cannot remove BlackMarks' embedded WM by excessive pruning while attaining a comparable accuracy with the baseline.

**Watermark Overwriting.** Assuming the attacker is aware of the watermarking methodology, he may attempt to corrupt the original WM by embedding a new one. In our experiments, we assume

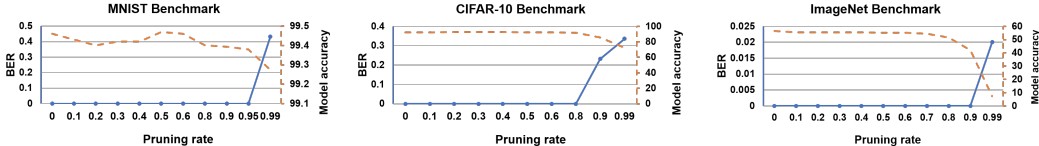

Figure 4: BlackMarks' robustness against parameter pruning. The key length is set to $K = 50$. The blue solid lines and orange dashed lines denote the BER and final test accuracy, respectively.

the adversary knows the targeted adversarial attack method employed by the model owner while the owner's signature and the key generation parameters remain secret. In this case, the attacker generates another set of WM keys with his own signature and key generation parameters to fine-tune the marked model following the steps outlined in Algorithm 1. Table 3 summarizes the accuracy of the overwritten DNN and the BER of the original WM signature ($K = 20, 30, 50$) for all three benchmarks. In our experiments, we assume the attacker uses the same key length as the owner to generate the new WM keys. BlackMarks can successfully extract the original WM in the overwritten DNN with zero BER, indicating its verifiability and robustness against WM overwriting attacks.

Table 3: BlackMarks' robustness against watermark overwriting attacks.

| Dataset | MNIST | | | CIFAR-10 | | | ImageNet (top-1) | | |
|---|---|---|---|---|---|---|---|---|---|
| Key Length | K=20 | K=30 | K=50 | K=20 | K=30 | K=50 | K=20 | K=30 | K=50 |
| Accuracy | 99.46% | 99.43% | 99.38% | 92.01% | 92.09% | 92.05% | 56.24% | 56.32% | 56.38% |
| BER | 0 | 0 | 0 | 0 | 0 | 0 | 0 | 0 | 0 |
| Detection Success | 1 | 1 | 1 | 1 | 1 | 1 | 1 | 1 | 1 |

## 5.3 SECURITY

The malicious adversary may try to find/design the exact WM key used by the model owner (*key collision*) and disturbs the WM extraction. The security of BlackMarks's WM key set is determined by the uncertainties involved in the key generation process (step 2 in Algorithm 1). Since the key generation parameters and the WM signature are assumed to be secret information provided by the owner as discussed in Section 4.1, even if the attacker is aware of the adversarial attack method used to generate the WM key, he cannot reproduce the exact same key due to the large searching space in the targeted adversarial attack method. Therefore, BlackMarks is secure against brute-force attacks.

## 5.4 INTEGRITY

Integrity requires that the watermarking technique shall not falsely claim the authorship of unmarked models. For multi-bit watermarking, such requirement means that if an unmarked model is queried by the owner's WM key set, the BER between the decoded signature from the model's predictions and the owner's signature shall not be zero. To evaluate the integrity of BlackMarks, we choose six unmarked models for each benchmark and summarize the results in Table 4. The first three models (M1-M3) have the same network topology but different weights as the watermarked model and the other three models (M4-M6) have different topologies as the marked model. For each benchmark, the owner queries these six unmarked models with her WM keys and tries to extract the signature. The computed BER is *non-zero* in all cases for three benchmarks, indicating that BlackMarks avoids claiming the ownership of unmarked DNNs and yields low false positive rates.

Table 4: Integrity evaluation of BlackMarks framework with key length $K = 50$.

| Dataset | MNIST | | | | | | CIFAR-10 | | | | | | ImageNet | | | | | |
|---|---|---|---|---|---|---|---|---|---|---|---|---|---|---|---|---|---|---|
| Unmarked Models | M1 | M2 | M3 | M4 | M5 | M6 | M1 | M2 | M3 | M4 | M5 | M6 | M1 | M2 | M3 | M4 | M5 | M6 |
| BER | 0.26 | 0.16 | 0.16 | 0.08 | 0.08 | 0.1 | 0.58 | 0.28 | 0.20 | 0.64 | 0.72 | 0.14 | 0.46 | 0.38 | 0.44 | 0.02 | 0.3 | 0.28 |

## 5.5 CAPACITY

One apparent advantage of BlackMarks over existing zero-bit black-box watermarking methods is its higher capacity as we discuss in Section 2. To further improve the capacity of the WM, BlackMarks can be easily generalized to embed more complex signatures instead of binary vectors. The amount of information carried by the owner's WM signature can be measured by *entropy* (Jaynes, 1957). More generally, if the owner specifies her signature (a numeric vector) with base $B$ and length $K$, the corresponding entropy can be computed as:

$$H = K \cdot log_2 B \qquad (2)$$

As can be seen from Eq. (2), a longer signature with a larger base value carries more information. Since we use a binary vector ($B = 2$) as the WM signature in this paper, the entropy can be simplified as $H = K$. To extend BlackMarks framework for embedding a base-$N$ signature, the owner needs to set the number of components in $K\_Means\_Clustering$ to $N$ (Algorithm 1) and change the encoding as well as decoding scheme of predictions correspondingly. BlackMarks is the first generic multi-bit watermarking framework that possesses high capacity in the black-box setting.

### 5.6 OVERHEAD

The WM extraction overhead is discussed in Section 4.3. Here, we analyze the runtime overhead incurred by WM embedding. Recall that the WM is inserted in the model by one-time fine-tuning of the target DNN with the regularized loss shown in Eq. (1). As such, the computation overhead to embed a WM is determined by computing the additive loss term $\mathcal{L}_{WM}$ during DNN training. BlackMarks has no communication overhead for WM embedding since the embedding process is performed locally by the model owner. To quantify the computation overhead for WM embedding, we measure the normalized runtime time ratio of fine-tuning the pre-trained model with the WM-specific loss and the time of training the original DNN from scratch. To embed the WM, we use the entire training data for MNIST and CIFAR-10 benchmark and $10\%$ of the training data for ImageNet benchmark in our experiments. The selected training data is concatenated with the WM key set to fine-tune the model. The results are visualized in Figure 5, showing that BlackMarks incurs a reasonable additional overhead for WM embedding (as low as 2.054%) even for large benchmarks.

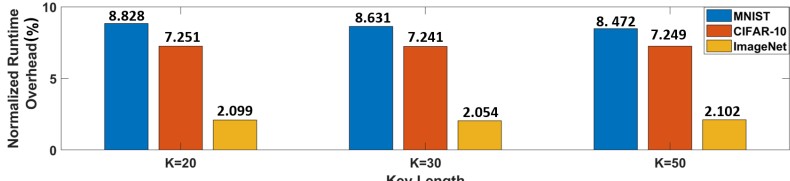

Figure 5: Normalized runtime ratio of BlackMarks' WM embedding with different key lengths.

## 6 DISCUSSION

Recall that WM embedding leverages a similar approach as 'adversarial training' while incorporating a WM-specific regularization loss (Section 4.1). Here, we study the effect of WM embedding on the model's robustness against adversarial attacks. Table 8 in Appendix A.3 compares the robustness of the pre-trained unmarked model and the corresponding watermarked model ($K = 50$) against different white-box adversarial attacks. It can be seen that for each type of the attack, the marked model has higher accuracy on the adversarial samples compared to the unmarked baseline. Such improvement is intuitive to understand since during WM embedding, the first term (cross-entropy loss) in the total regularized loss (see Eq. (1)) enforces the model to learn the correct predictions on training data as well as on the WM keys ('adversarial samples'), thus having a similar effect as 'adversarial training' (Kurakin et al., 2016). Therefore, BlackMarks has a side benefit of improving the model's robustness against adversarial attacks.

In the future, we plan to extend BlackMarks framework to the multi-user setting for fingerprinting purpose. Chen et al. (2018) present the first collusion-resilient DNN fingerprinting approach for unique user tracking in the white-box setting. To the best of our knowledge, no black-box fingerprinting has been proposed due to the lack of black-box multi-bit watermarking schemes. BlackMarks proves the feasibility of black-box fingerprinting methods and builds the technical foundation.

## 7 CONCLUSION

We propose BlackMarks, the first black-box multi-bit watermarking framework for IP protection of DNNs. To the best of our knowledge, this work provides the first empirical evidence that embedding and extracting multi-bit information using the model's predictions are possible. Our comprehensive evaluation of BlackMarks' performance on various benchmarks corroborates that BlackMarks coherently embeds robust watermarks in the output predictions of the target DNN with an additional overhead as low as $2.054\%$. BlackMarks possesses superior capacity compared to all existing zero-bit watermarking techniques and paves the way for future black-box fingerprinting techniques.

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

APPENDIX

A.1 EXPERIMENTAL SETUPS

**Key Generation Method.** For WM key generation, we use JSMA (Papernot et al., 2016b) for MNIST and CIFAR-10 benchmark, and MIM (Dong et al. 2018) for ImageNet benchmark. We do not employ MIM for MNIST and CIFAR-10 benchmark since gradient-based MIM provides a limited amount of randomness during key generation when the total number of categories in the dataset is small ($C = 10$), making the resulting WM keys vulnerable to over-writing attacks. JSMA is used to generate adversarial samples as the WM key images in this case, which works by searching for pixel pairs to perturb and introduces sufficient uncertainties.

For ImageNet dataset (where the total number of categories is $C = 1000$), the sizes of the code-bit cluster '0' and cluster '1' are larger than ones in MNIST and CIFAR-10 dataset. Therefore, the searching space for targeted adversarial samples is larger and the probability of *WM key collision* is smaller, ensuring the robustness of the generated WM keys against the WM over-writing attack. JSMA is not applied to the ImageNet benchmark since the excessive memory requirement (Xu et al., 2017) cannot be satisfied by our 11.74GiB test machine.

**Model Fine-tuning for WM Embedding.** To embed the WM, we set the hyper-parameter $\lambda$ to $0.5$ for MNIST and CIFAR-10 benchmark, and to $0.01$ for ImageNet benchmark in our experiments. The pre-trained unmarked model is fine-tuned for 15 epochs with the regularized loss in Eq. (1) for all benchmarks. We use the same batch size and the optimizer setting used for training the original neural network, except that the learning rate is reduced by a factor of 10. Such retraining procedure coherently encodes the WM key in the distribution of output activations while preventing the accuracy drop on the legitimate data.

## A.2 NETWORK ARCHITECTURES

We summarize the DNN topologies used in each benchmark and the corresponding WM embedding results in Table 5. Here, $K$ denotes the size of the WM key set, which is also equal to the length of the owner's WM signature.

Table 5: Benchmark network architectures. Here, $64C3(1)$ indicates a convolutional layer with $64$ output channels and $3 \times 3$ filters applied with a stride of 1, $MP2(1)$ denotes a max-pooling layer over regions of size $2 \times 2$ and stride of 1, and $512FC$ is a fully-connected layer with $512$ output neurons. $BN$ denotes batch-normalization layer. ReLU is used as the activation function.

| Dataset | Baseline Accuracy | Accuracy of Marked Model | | | DL Model Architecture |
|---|---|---|---|---|---|
| MNIST | 99.44% | K = 20 | K = 30 | K = 50 | 1*28*28-32C3(1)-BN-32C3(1)-MP2(1)-BN |
| | | 99.44% | 99.46% | 99.42% | -64C3(1)-BN-Flatten-BN-512FC-BN-10FC |
| CIFAR10 | 92.09% | K = 20 | K = 30 | K = 50 | Please refer to Zagoruyko & Komodakis (2016). |
| | | 92.10% | 92.28% | 92.19% | |
| ImageNet (top-1) | 56.39% | K = 20 | K =30 | K =50 | Please refer to Krizhevsky et al. (2012). |
| | | 56.28% | 56.31% | 56.36% | |

## A.3 ADDITIONAL EXPERIMENTAL RESULTS

In this section, we provide supplementary experimental results to support the evaluation of Black-Marks' performance in the paper.

**Robustness against Model Fine-tuning Attack.** Table 6 shows the effect of model fine-tuning on BlackMarks WM extraction as discussed in Section 5.2. The BER remains zero after the marked model is fine-tuned for various numbers of epochs, suggesting that BlackMarks is robust against fine-tuning attacks. Note that the number of fine-tuning epochs for ImageNet benchmark is smaller than the other two since the number of epochs needed to train the ImageNet benchmark from scratch is 70 whereas the other benchmarks take around 200 epochs to be trained.

Table 6: BlackMarks' robustness against fine-tuning attacks. The key length is set to $K = 50$.

| Dataset | MNIST | | | CIFAR-10 | | | ImageNet (top-1) | | |
|---|---|---|---|---|---|---|---|---|---|
| # Fine-tuning Epochs | 20 | 50 | 100 | 20 | 50 | 100 | 5 | 10 | 20 |
| Accuracy | 99.46% | 99.48% | 99.51% | 92.34% | 92.36% | 92.40% | 55.19% | 55.13% | 55.17% |
| BER | 0 | 0 | 0 | 0 | 0 | 0 | 0 | 0 | 0 |
| Detection Success | 1 | 1 | 1 | 1 | 1 | 1 | 1 | 1 | 1 |

**Evaluation of BlackMarks' Integrity.** Table 7 presents the results of the integrity evaluation of BlackMarks when different WM key sizes are used. One can see that the BER between extracted signature from the unmarked model and the owner's true signature is *non-zero* in all cases, indicating that BlackMarks respects the integrity requirement with various key lengths.

**Effect of BlackMarks' WM Embedding on Model's Robustness.** Table 8 compares the robustness of the pre-trained unmarked model and the corresponding marked model against various white-box adversarial attacks as discussed in Section 6. It can be observed that the watermarked model has higher classification accuracy on the adversarial samples compared to the unmarked model,

Table 7: Integrity evaluation of BlackMarks framework with various key lengths.

| Dataset | | MNIST | | | | | | CIFAR-10 | | | | | | ImageNet | | | | | |
|---|---|---|---|---|---|---|---|---|---|---|---|---|---|---|---|---|---|---|---|
| Unmarked Models | | M1 | M2 | M3 | M4 | M5 | M6 | M1 | M2 | M3 | M4 | M5 | M6 | M1 | M2 | M3 | M4 | M5 | M6 |
| BER | K=20 | 0.25 | 0.2 | 0.25 | 0.05 | 0.05 | 0.15 | 0.15 | 0.15 | 0.05 | 0.35 | 0.75 | 0.15 | 0.50 | 0.45 | 0.55 | 0.25 | 0.20 | 0.25 |
| | K=30 | 0.27 | 0.23 | 0.23 | 0.03 | 0.03 | 0.1 | 0.15 | 0.15 | 0.05 | 0.35 | 0.75 | 0.2 | 0.37 | 0.27 | 0.33 | 0.1 | 0.2 | 0.17 |
| | K=50 | 0.26 | 0.16 | 0.16 | 0.08 | 0.08 | 0.1 | 0.58 | 0.28 | 0.20 | 0.64 | 0.72 | 0.14 | 0.46 | 0.38 | 0.44 | 0.02 | 0.3 | 0.28 |

indicating that BlackMarks WM embedding process has a positive impact on the model's robustness against adversarial attacks.

Table 8: Robustness comparison between unmarked and marked models against adversarial attacks.

| Accuracy on Adversarial Set | MNIST | | CIFAR-10 | |
|---|---|---|---|---|
| | Unmarked Model | Marked Model | Unmarked Model | Marked Model |
| **FGSM** (untargeted, Madry et al., 2017) | 58.67% | 75.03% | 36.51% | 36.93% |
| **JSMA** (targeted, Papernot et al., 2016b) | 4.98% | 10.20% | 76.56% | 77.21% |
| **MIM** (targeted, Dong et al., 2018) | 59.24% | 77.47% | 53.97% | 56.82% |
| **CW** (targeted, Carlini & Wagner, 2017) | 87.50% | 89.91% | 43.03% | 44.84% |

