# OpenReview forum: "BlackMarks: Black-box Multi-bit Watermarking for Deep Neural Networks"
_ICLR.cc/2019/Conference_

### Official Review · AnonReviewer1 · 2018-11-01

**Rating:** 4
**Confidence:** 4

**Review:**

Strengths:

Well written paper, covers most of the relevant related work
Technique is conceptually easy to understand (~ adversarial training)

Weaknesses:

Unclear set of desiderata properties for a watermarking technique
No formal guarantees are verified, the mechanism is only tested
Attacks tested are not tailored to the technique proposed

Feedback and rebuttal questions:

This submission is easy to read and follow, and motivates the problem of watermarking well in light of intellectual property concerns. The technique proposed exploits unused capacity in the model to train it to associate specific inputs (computed adversarially) to specific outputs (the keys). Watermarking succeeds when the bit error rate between the predicted signature and the expected one is zero. This approach is conceptually easy to understand.

The experimental setup used to evaluate the approach is however limited. First, it is unclear why desiderata stated in Section 3.1 and summarized in Table 1 are necessary and sufficient. Would you be able to justify their choice in your rebuttal? For instance, the “security” requirement in Table 1 overlaps with “fidelity”. Similarly, the property named “integrity” really refers to only a subset of what one would typically describe as integrity. It basically calls for a low false positive or high precision.

The attack model described in Section 3.2 only considers three existing attacks: model fine-tuning, parameter pruning and watermark overwriting. These attacks do not consider how the adversary could adapt and they are not optimal strategies for attacking the specific defensive mechanism put in place here. For instance, could you explain in your rebuttal why pruning the smallest weights in the architecture in the final architecture would help with removing adversarial examples injected to watermark the model? Similarly, given that adversarial subspaces have large volumes, it makes sense that multiple watermarks could be inserted simultaneously and thus watermark overwriting attacks would fail.

If the approach is based on exploring unused capacity in the model, the adversary could in fact attempt to use a compression technique to preserve the model’s behavior on the task and remove the watermarking logic. For instance, the adversary could use an unlabeled set of inputs and have them labeled by the watermarked model. Because these inputs will not be “adversarial”, the watermarked model’s decision surface used to encode the signatures will remain unexplored during knowledge transfer and the resulted compressed or distilled model would solve the original task without being watermarked. Is this an attack you have considered in your experiments and if not could you elaborate why one may exclude it in your rebuttal?

Minor comments:

P3: Typo “Verifiabiity”
P5: Could you add a reference or additional experimental results that justify why transferable keys would be located near the decision boundaries?

---

### Official Review · AnonReviewer2 · 2018-11-05
**Some interesting ideas, but a better evaluation is needed to show the effectiveness of the method**

**Rating:** 4
**Confidence:** 4

**Review:**

summary:

The paper proposes an approach for model watermarking (i.e., watermarking a trained neural neteowrk). The watermark is a bit string, which is embedded in the model as part of a fine-tuning procedure, and can be decoded from the network from the model's specific predictions for a specific set of inputs (called keys) chosen during the fine-tuning step. The process generates a watermark when we can be confident that a model that didn't go through the exact same fine-tuning procedure gives significantly different predictions on the set of keys. The application scenario is when a company A wants to deploy a model for which A has IP ownership, and A wants to assess whether a competitor is (illegaly) re-using A's model. The approach presented in the paper works in the black-box setting, meaning that whether a model posesses the watermark can be assessed only by querying the model (i.e., without access to the internals of the model).

The overall approach closely follows Merrer et al. (2017), but extends this previous work to multi-bit watermarking. The similarity with Merrer et al. is that keys are generated with a procesdure to generate adversarial examples, watermarking is performed by specifically training the network to give the source label (i.e., the label of the image from which the adversarial example has been generated). The differences with Merrer et al. lie in the fact that each key encoded a specific bit (0 or 1), and the multi-bit watermark is encoded in the predictions for all keys (in case of a multi-class classifier, the labels are first partitionned into two clusters to map each class to either 0 or 1). In contrast, Merrer et al. focused on "zero-bit" watermarking, meaning that all keys together are only used to perform a test of whether the model has been watermarked (not encode the watermark). Another noticeable difference with Merrer et al. is in step 4 of the algorithm, in which several unmarked models are generated to select better key images.

comments:

While overall the approach makes sense and most of the design decisions seem appropriate, many questions are only partly addressed. My main concerns are:
1- the watermarks are encoded in adversarial examples for which the trained model gives the "true" label (i.e., the watermark is embedded in adversarial examples on which the model is robust). The evaluation does not address the concerns of false alarms on models trained to be robust to adversarial examples. Previous work (e.g., Merrer et al.) study at least the effect of fine-tuning with adversarial examples..

2- A watermark of length K is encoded in K images, and the test for watermarking is "The owner can prove the authorship of the model if the BER is zero.". This leaves little room to model manipulation. For instance, the competitor could randomize its predictions once in a while (typically output a random label for one out of K inputs), with very small decrease in accuracy and yet would have a non-negligible probability of having a non-zero BER.

other comments:
1- overhead section: in step 4 of the algorithm, there is a mention of "construct T unmarked models": why aren't they considered in the overhead? This seems to be an extremely significant part of the cost (the overall cost seems to be more T times the cost of building a single unmarked model rather than a few percent)

2- step 2 page 4: "The intuition here is that we want to filter out the highly transferable WM keys": I must have misunderstood something here. Why are highly transferable adversarial examples a problem? That would be the opposite: if we want the key to generate few false alarms (i.e., we do not want to claim ownership of a non-watermarked model), then we need the adversarial examples to "transfer" (i.e., be adversarial for non-watermarked models), since the watermarked model predicts the source class for the key. Merrer et al. (2017) on the contrary claim " As such adversaries seem to generalize across models [...] , this frontier tweaking should resist model manipulation and yield only few false positives (wrong identification of non marked model).", which means that transferability of adversarial examples is a fundamental assumption underlying the approach.

3- under Eq. 1: "Note that without the additional regularization loss (LWM), this retraining procedure resembles ‘adversarial training’ (Kurakin et al., 2016).": I do not understand that sentence. Without L_{WM}, the loss is the usual classification loss (L_0), and has nothing to do with adversarial training.

4- more generally, the contribution of the paper is on multi-bit watermarking, but there is no clear application scenario/experiment  where the multi-bit is more useful than the zero-bit watermarking.

---

### Official Review · AnonReviewer3 · 2018-11-06

**Rating:** 5
**Confidence:** 3

**Review:**

A method for multi-bit watermarking of neural networks in a black-box setting is proposed. In particular, the authors demonstrate that the predictions of existing models can carry a multi-bit string that can later be used to verify ownership.
Experiments on MNIST, CIFAR-10 and ImageNet are presented in addition to a robustness assessment w.r.t. different WM removal attacks.

Questions/Comments:

Regarding the encoding scheme, a question that came up is whether one needs to perform clustering on the last layer before the softmax? In principle, this could be done at any point, right?

Another question is how the method scales with the key length. Did you experiment with large/small values of K (e.g., 100,200,...)? It would be interesting, e.g., to see a plot that shows key length vs. accuracy of the marked model, or, key
length vs. detection success (or BER).

Apart from these comments, how does the proposed model compare to zero-bit WM schemes? I am missing a clear comparison to other, related work, as part of the experiments. While there might not exist other "black-box multi-bit"
schemes in the literature, one could still compare against non-multi-bit schemes.

In light of a missing comparison, my assessment is "Marginally below acceptance threshold", but I am willing to vote
this up, given an appropriate response.

---

### Meta-Review · Area_Chair1 · 2018-12-17
**reject**

**Confidence:** 4
**Recommendation:** Reject

**Metareview:**

The reviews agree the paper is not ready for publication at ICLR.